# Finding the Optimal Fatty Acid Composition for Biodiesel Improving the Emissions of a One-Cylinder Diesel Generator

Rafael R. Maes [1,*], Geert Potters [1,2], Erik Fransen [3], Fátima Calderay Cayetano [4], Rowan Van Schaeren [1] and Silvia Lenaerts [2]

1   Antwerp Maritime Academy, Noordkasteel Oost 6, 2030 Antwerp, Belgium; geert.potters@hzs.be (G.P.); rowan.van.schaeren@hzs.be (R.V.S.)
2   Department of Bioscience Engineering, University of Antwerp, 2020 Antwerp, Belgium; silvia.lenaerts@uantwerpen.be
3   STATUA Center for Statistics, University of Antwerp, 2610 Antwerp, Belgium; erik.fransen@uantwerpen.be
4   Departamento de Máquinas y Motores Térmicos, Universidad de Cadiz, 11001 Cadiz, Spain; fatima.calderay@uca.es
*   Correspondence: raf.maes@hzs.be

**Abstract:** Nitrogen oxides ($NO_x$) and particulate matter (PM) currently are the main pollutants emitted by diesel engines. While there is a start in using hybrid and electric cars, ships will still be fueled by mineral oil products. In the quest to achieve zero-pollution and carbon-free shipping, alternative forms of energy carriers must be found to replace the commonly used mineral oil products. One of the possible alternative fuels is biodiesel. This paper explores the optimization of the composition of biodiesel in order to reduce the concentration of particulate matter and $NO_x$ in exhaust gases of a one-cylinder diesel generator.

**Keywords:** particulate matter; $NO_x$; biodiesel; fatty acid composition

## 1. Introduction

Considering the current use of natural oil reserves, a huge number of these reserves are used for transport. As our natural resources of oil and gas are limited, we need to look for other means to produce energy. Moreover, the way our energy resources are used is equally unsustainable in view of pollution. Alternative energy sources are not only needed to prevent too much pollution but are a "must" to a sustainable future. Two of the most widespread alternative energy sources are wind and solar energy. The energy consumption of road transport is nowadays focused on electrification. However, marine and air transport show such high energy consumption that electrification will not take place in the near future. Biodiesel could provide a solution for the large demand for fuel consumption. Currently, there is still a huge amount of mineral fuel used in these transport modes. A change is necessary to control environmental pollution.

The use of biodiesel is a solution which deals with the diminishing amount of mineral diesel and also reduces the amount of greenhouse gases. The goal of this study was to look for a method to calculate the optimal fatty acid composition of a biodiesel in order to have as low as possible concentrations of PM and $NO_x$ in the exhaust gases of a one-cylinder diesel engine.

$CO_2$ emissions are an environmental problem, but particulate matter (PM) and $NO_x$ also pose serious environmental and health risks [1] and are currently under even more scrutiny. Decreasing the $NO_x$ and Particulate Matter (PM) emissions could lead to a new and cleaner alternative for mineral diesel. The effect of forming $NO_x$ and PM is, however, a function of the different fatty acids found in biodiesel. Kalligeros et al. showed that blends of biodiesel, produced from olive oil and sunflower oil, and marine diesel up to 50% show a decrease in emissions [2]. A decrease in HC, CO, PM and $NO_x$ was measured in contrast to

a slight increase in fuel consumption. Blends of diesel and soy biodiesel (B0, B10, B20, B50) were tested as well and, again, a decrease in PM concentration was measured [3]. The same result was found in [4] using biodiesel produced from palm oil. Blends (B5, B10, B20 and B100) were tested on PM and a decrease in PM concentration was found. However, not only lower emissions prove to be an advantage, but also the biodegradability and lubricity of biodiesel [5].

Biodiesel is a mixture of different esters of fatty acids, and each fatty acid has, due to its molecular structure, a different combustion characteristic, and thus a different way of forming particulate matter (PM) and nitrogen oxides (NO and $NO_2$) [6]. Locating an optimal fatty acid composition in order to have the lowest possible PM and $NO_x$ concentration in the exhaust gases would lead to future generations not only profiting from endless quantities of biodiesel, but at the same time enjoying improved air quality. This article demonstrates that it is indeed possible to find such an Optimal Fatty Acid Composition (OFAC).

The most common fatty acids found in vegetable oils are palmitic acid, stearic acid, oleic acid, linoleic acid and linolenic acid, which can be turned into methyl esters (Fatty Acid Methyl Esters, FAME). Each FAME has its own combustion characteristic, meaning that they all produce $NO_x$ and PM in a different way during combustion.

Another advantage of using biodiesel instead of mineral diesel is the positive effect on air quality. Biodiesel can be produced in an almost 100% renewable way [1,7], depending on the use of alcohol or bioalcohol in the synthesis. Combustion of biodiesel has a positive effect on both net PM and $NO_x$ emissions and net $CO_2$ emissions. The greenhouse effect could be limited by using biodiesel with an OFAC. The production of $NO_x$ and PM is influenced by the amount or ratio of different fatty acids found in the biodiesel. For example, blends of up to 50% of biodiesel produced from olive oil and sunflower oil with marine diesel show a decrease in emissions [2]. A decrease in CO, PM and $NO_x$ was measured in contrast to a slight increase in fuel consumption. Similar decreases of PM emissions were found when blends of diesel and soy biodiesel (B0, B10, B20, B50) [3] or biodiesel produced from palm oil were used (blends B5, B10, B20 and B100) [4].

The major objective of this research was therefore to find an optimized mixture of these FAMEs in order to minimize NO, $NO_2$ and PM. Such an optimal fatty acid composition could indeed support a sustainable production of biodiesel as well as improve air quality. This article presents a methodology to find this optimal fatty acid composition.

## 2. Materials and Methods

### 2.1. Scheme of the Emission Measurement

PM was measured by using a DustTrak DRX model 8533 (TSI Instruments, Wycombe, UK). The sampling time was set at 60 s and the sampling period at 10 s. NO and $NO_2$ were measured by a Crown Con Gas-Pro (Crowncon Detection Instruments Ltd., Abingdon, UK). NO and $NO_2$ were measured three times per minute. The measuring range for NO is 0–100 ppm and for $NO_2$ it is 0.0–20.0 ppm. The measuring range for PM is 0.001–150 mg/m$^3$. In order to reduce the concentrations of NO, $NO_2$ and PM within the measurement range of the measuring devices, the samples of the exhaust gases were diluted by a factor of 6.1 ± 0.2.

Biodiesel is produced by transesterification of oils with (usually) methanol, thus becoming a mixture of fatty acid methyl ester. The most common fatty acids found in vegetable oils are palmitic acid, stearic acid, oleic acid, linoleic acid and linolenic acid, which can be turned into FAME [8]. Biodiesel conform EN 14214 was provided and pure methyl esters of these five fatty acids, purchased at TCI Europe, were added at varying concentrations. The fatty acid methyl ester composition of the biodiesel was analyzed at Vlaamse Instelling voor Technologisch Onderzoek (VITO) using gas chromatography and values are given in mass percentage (Table 1).

**Table 1.** Results of the fatty acid composition of the different biodiesels in mass percentage.

| RME (m%) | C16_0 | C18_0 | C18_1 | C18_2 | C18_3 |
|----------|-------|-------|-------|-------|-------|
| AW | 11.3 | 1.8 | 58 | 20.4 | 8.6 |
| BW | 11.8 | 1.8 | 57.7 | 20.2 | 8.5 |
| CW | 6.4 | 1.5 | 60.4 | 21.9 | 9.7 |
| DW | 13 | 1.9 | 57.1 | 19.8 | 8.3 |
| EW | 12.6 | 1.7 | 56.7 | 21.1 | 8.1 |
| FW | 12.2 | 1.8 | 56.9 | 21.1 | 8.1 |
| GW | 7.1 | 1.5 | 59.3 | 23 | 9.1 |
| HW | 11.2 | 2.2 | 57.2 | 21 | 8.4 |

The effect on the concentration of $NO_x$ and PM in the exhaust gas of different fatty acid compositions of this biodiesel was tested on a one-cylinder diesel generator without turbo mechanism. As previously stated, the goal was to see if there was a significant influence in producing $NO_x$ and PM due to the different composition of FAME in the biodiesel, and if there was, whether a method to optimize the fatty acid composition could be found. The scheme of the emission measurement is shown in Figure 1.

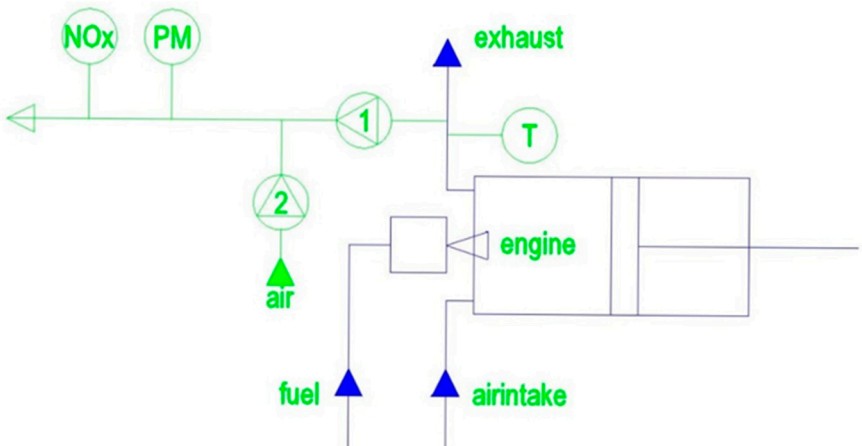

**Figure 1.** Scheme of the experimental set-up. 1 is the sampling compressor, 2 is the dilution compressor and T is the temperature sensor. PM is the particulate matter sensor and $NO_x$ is the sensor measuring NO and $NO_2$.

*2.2. Engine*

The experiments were conducted using a four-stroke one-cylinder diesel generator (Table 2).

**Table 2.** Engine properties of the JavacNanomag NM 7500 B (KM 186 FA) engine.

| JavacNanomag NM 7500 B (KM 186 FA) | |
|---|---|
| Injection system | Direct injection |
| Type | Single cylinder |
| Cooling | Air cooled |
| Aspiration | Naturally aspirated |
| Bore (mm) | 86 |
| Stroke (mm) | 70 |
| Compression ratio | 19 |

In order to test the different types of fuel samples, the generator was equipped with a reservoir that could be cleaned after each measurement had been run. The measurements were done on a three-phase system. Three identical hot air blowers (Remington REM 2 ECA) were used as load. The load could be increased in steps of 0.7 kW.

### 2.3. Measuring Protocol

The engine load was increased in steps of 0.7 kW, and at each different level of power, the measurement only took place once the temperature of the exhaust gas was stable and did not exceed differences of more than a few tenths of a degree. At level of load 0 kW (idle), 0.7 kW, 1.4 kW, 2.1 kW, 2.8 kW, 3.5 kW, 4.2 kW, 4.9 kW and 5.6 kW, PM, NO and $NO_2$ concentrations were measured. The contents of $NO_x$ and PM were measured in a tube in which a constant flow of exhaust gas was taken by means of a sampling compressor and, subsequently, this constant flow was diluted by adding a constant flow of ambient air. To reach this goal, an air compressor at constant rotation speed was used (Figure 1), which resulted in a dilution of $6.1 \pm 0.2$. After dilution, $NO_x$ and PM concentrations were determined, to ensure that concentrations did not surpass the measuring limits of the measuring devices. This dilution factor does not affect the analysis, because the relative values of the different concentrations remain unchanged.

### 2.4. Fatty Acid Composition Analysis

Analysis of the specific concentrations of the five components was done at VITO and the results given in relative values (Table 2). Fatty acid composition was determined using gas chromatography coupled to a flame ionisation detector (GC-FID). The GC was equipped with a ZB-WAX column (30 m × 0.25 mm × 0.25 μm). The oven was programmed as follows: 75 °C for 1 min, ramp at 25 °C/min to 200 °C, ramp at 5 °C/min to 230 °C, hold for 4 min. The total runtime of the method was 16 min. The injector temp was set at 250 °C, and the FID was set at 320 °C. One μL was injected in split mode (1:10). Authentic reference standards were purchased for the five fatty acid methyl esters listed below and were used for calibration. Samples were diluted in toluene to approximately 5 μg/g (m/m) before analysis. Concentrations (Table 1) are given in mass percentages: C16_0 is methyl palmitate, C18_0 is methyl stearate, C18_1 is methyl oleate, C18_2 is methyl linoleate and C18_3 is methyl linolenate.

### 2.5. Calculations and Statistical Analysis

The eight mixtures of biodiesel to which fatty acid methyl esters were added at varying concentrations were tested at 9 different loads, ranging from 0 kW (idle) up to 5.6 kW. At each intensity of load, the PM, NO and $NO_2$ concentrations were recorded.

Modeling the relation between the concentration of the methyl esters in the different mixtures and the exhaust was carried out in two steps, to account for the effect of load on the exhaust, and to allow for possible nonlinear effects.

In the first step, residuals after regression on load were calculated. In brief, a linear regression model was fitted with the exhaust of interest (PM, NO or $NO_2$) as dependent (Y) variable, and the load (w) as independent variable. An overall model was fitted, allowing for a nonlinear relation between the exhaust and the load. We started from a fourth-order model. To avoid overfitting, this model was simplified using stepwise backward elimination.

Figure 2a–c shows the regression curves of the final model for NO, $NO_2$ and PM, respectively. Each dot corresponds to one mixture (measured at the load indicated on the *X*-axis), with some mixtures having an exhaust above the regression curve (positive residual) and others an exhaust below the overall curve (negative residual). By working with the residuals in the subsequent analysis, the effect of the load was corrected in the second regression. In this second step, we searched for a relation between the residual (=the load-corrected exhaust) and the methyl ester composition. This analysis is aimed at finding out to what extent the FAME determines whether an observation has an exhaust above or

below the overall regression line in Figure 2a–c. The residuals were entered as dependent variable in a regression model, with one of the FAMEs as independent variable. One separate regression model was fitted for each of the five additives, for the three exhausts. To model the relation between the residual (dependent variable), and the concentration of the additive, allowing for nonlinear (higher order) effects, and the model building started from a third-order polynomial regression model that was simplified by backward elimination. The regression coefficients of the final model are shown in Tables 3–5.

**Table 3.** Regression coefficients for load versus NO.

|  | Estimate | Std. Error | Pr(>∣t∣) |
| --- | --- | --- | --- |
| intercept | −28.179 | 7.628 | 0.000607 |
| kW | 21.332 | 6.596 | 0.000232 |
| I (kW$^2$) | −4.456 | 1.794 | 0.016897 |
| I (kW$^3$) | 0.354 | 0.156 | 0.027219 |

**Table 4.** Regression coefficients for load versus NO$_2$.

|  | Estimate | Std. Error | Pr(>∣t∣) |
| --- | --- | --- | --- |
| intercept | 1.385 | 0.058 | $<2 \times 10^{-16}$ |
| kW | 1.346 | 0.096 | $<2 \times 10^{-16}$ |
| I (kW$^2$) | −0.529 | 0.041 | $<2 \times 10^{-16}$ |
| I (kW$^3$) | 0.045 | 0.005 | $9.95 \times 10^{-14}$ |

**Table 5.** Regression coefficients for load versus PM.

|  | Estimate | Std. Error | Pr(>∣t∣) |
| --- | --- | --- | --- |
| intercept | 0.453 | 0.097 | $1.54 \times 10^{-5}$ |
| kW | 0.515 | 0.081 | $1.91 \times 10^{-8}$ |
| I (kW$^2$) | −0.069 | 0.014 | $4.99 \times 10^{-6}$ |
| I (kW$^3$) |  |  |  |

To find the optimal values, the zeroes of the first derivative were calculated. The constraint is that the sum of the five components has to be 100%.

Once the model, a polynomial, for every FAME was retrieved by statistical analysis, the optimal points were calculated by putting the first derivative to zero. If a minimal value is found, the concentration of this FAME produces the lowest concentration of NO$_x$ or, respectively, PM.

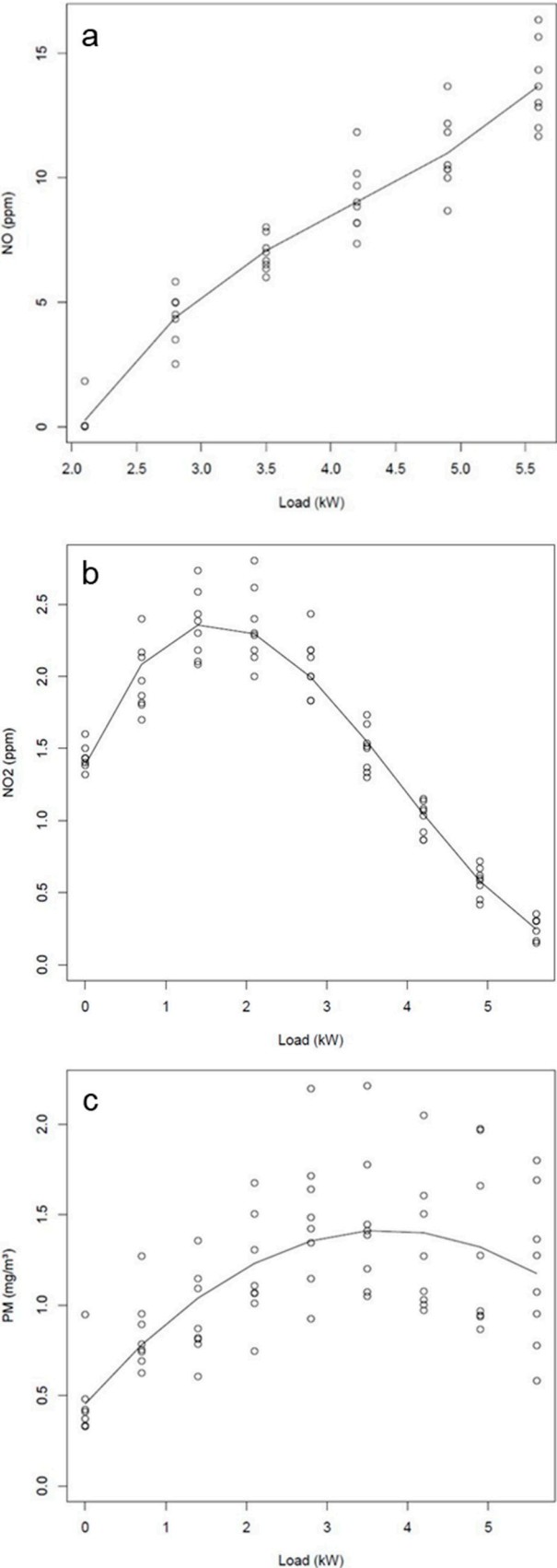

**Figure 2.** (**a**): Regression curve load versus NO; (**b**): regression curve load versus $NO_2$; (**c**): regression curve load versus PM.

## 3. Results

The main aim of this paper was to construct a statistical method to find an optimal fatty acid concentration for RME, leading to a minimal emission of $NO_x$ and PM. To search for an optimal concentration of FAMEs in biodiesel that minimizes the exhaust of NO, $NO_2$ and PM, several mixtures of biodiesel, characterized by varying concentrations of the five most common FAMEs, were tested. As outlined in the measuring protocol section, the engine ran at different load levels, and the exhaust concentrations of NO, $NO_2$ and PM were registered.

To model the relation between the concentration of the additive and the exhaust, accounting for the effect of power intensity, polynomial regression models were fitted as described in the statistical analysis. In brief, the residuals after regression on the power were modelled versus the concentrations of each FAME.

The regression coefficients for these models and the significance levels are shown in Tables 3–5. The relation between the residuals and the concentration of the additives, for each of the exhausts, are shown in Figures 3–5.

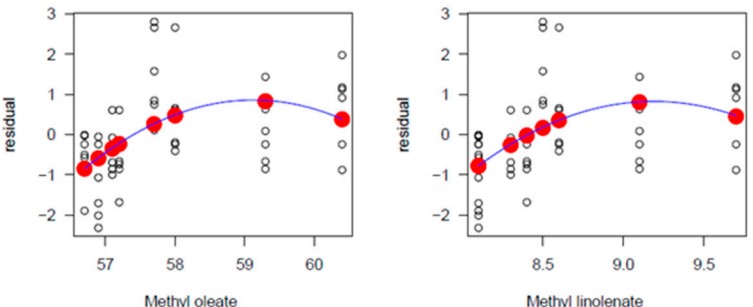

**Figure 3.** Relation between residuals and concentration in mass percentage of the particular fatty acid.

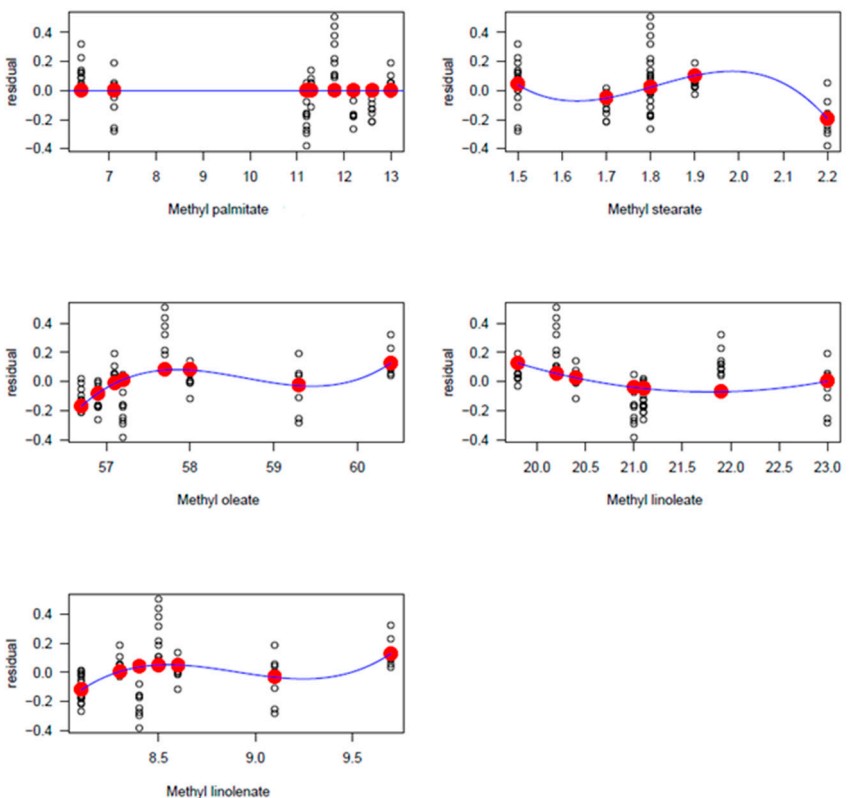

**Figure 4.** Relation between residuals and concentration in mass percentage of the particular fatty acid.

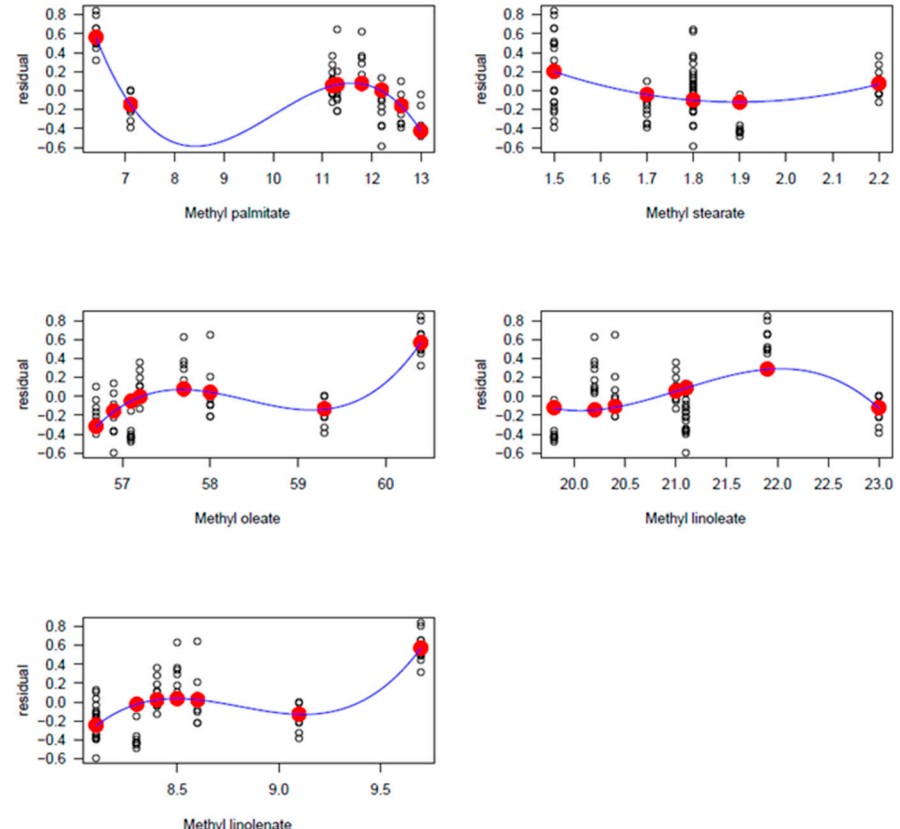

**Figure 5.** Relation between residuals and concentration in mass percentage of the particular fatty acid.

The results of the statistical analysis can be found in the underlying tables, together with the calculated values of the concentrations. Values in a yellow accent have been calculated using the constraint that the total sum of the concentrations has to be 100%.

The values represent the results from the calculation of the extreme values of the polynomial. Min and max are the calculated concentrations of the FAME that result in the minimal and, respectively, maximal values of the polynomial. The value indicated in yellow represents a concentration obtained using the constraint that the sum of all percentages has to add up to 100%.

The *Y*-axis of Figures 3–5 represents the residuals after regression, modeling the exhaust versus the load. This was performed to correct exhaust values for the load of the experiment. Subsequently, these load-corrected exhaust values are plotted versus the concentration of the FAME.

### 3.1. Results for NO

NO exhaust showed a linear relation with C18_1 and C18_3, while the relation with the other FAME concentrations was not significant (Table 6; not-significant values were indicated with "ns"). No specific FAME blend can be found that produces a minimal concentration of NO during combustion.

**Table 6.** Regression coefficients for NO content produced by each specific fatty acid in the exhaust gases. ns: not statistically significant.

| | **Regression Coefficients** | | | | | |
| | **Intercept** | **Linear** | **Square** | **Cube** | **Min** | **Max** |
|---|---|---|---|---|---|---|
| C16_0 | ns | ns | ns | ns | | |
| C18_0 | ns | ns | ns | ns | | |
| C18_1 | −1013.75 | 34.32077 | −0.29024 | ns | | 59.13 |
| C18_2 | ns | ns | ns | ns | | |
| C18_3 | −113.216 | 24.82194 | −1.3507 | ns | | 9.19 |

The Y-value of the observation represents the residual of the exhaust, after regression on the load intensity. The *X*-axis shows the concentration of the particular fatty acid in the mixture. The solid line represents the fitted polynomial regression model, and the red dots show the predicted values according to the regression model. These results are for NO concentrations for each significant FAME.

### 3.2. Results for $NO_2$

$NO_2$ exhaust showed a third-order relation with all FAMES except C16_0. Based upon the regression coefficients, an optimal composition for minimal $NO_2$ exhaust was calculated, as shown in Table 7 (not-significant values were indicated with "ns").

**Table 7.** Regression coefficients for $NO_2$ content produced by each specific fatty acid in the exhaust gases. ns: not statistically significant.

| | **Regression Coefficients** | | | | | |
| | **Intercept** | **Linear** | **Square** | **Cubic** | **Min** | **Max** |
|---|---|---|---|---|---|---|
| C16_0 | ns | ns | ns | ns | 7.03 | |
| C18_0 | 54.51865 | −92.1089 | 51.39976 | −9.47114 | 1.63 | 1.98 |
| C18_1 | −10856.3 | 555.4831 | −9.47227 | 0.053831 | 60.32 | 57.06 |
| C18_2 | 23.66178 | −2.17834 | 0.049984 | ns | 21.78 | |
| C18_3 | −400.251 | 135.3405 | −15.2309 | 0.570473 | 9.24 | 8.56 |

Once again, the values represent the results from the calculation of the extreme values of the polynomial. Min and max are the calculated concentrations of the FAME that result in the minimal and, respectively, maximal values of the polynomial. The highlighted value represents a concentration obtained using the constraint that the sum of all percentages has to add up to 100%. The minimal and maximal values of the polynomial were calculated. The value of C16_0 is calculated by using the constraint that the sum of the relative values has to be 100%.

### 3.3. Results for PM

Once again, the values represent the results from the calculation of the extreme values of the polynomial (Table 8; not-significant values were indicated with "ns"). Min and max are the calculated concentrations of the FAME that result in the minimal and respectively maximal values of the polynomial. The highlighted value represents a concentration obtained using the constraint that the sum of all percentages has to add up to 100%. The minimal and maximal values of the polynomial were calculated. The value of C16_0 is calculated by using the constraint that the sum of the relative values has to be 100%.

**Table 8.** Regression coefficients for PM content produced by each specific fatty acid in the exhaust gases. ns: not statistically significant.

| | Regression Coefficients | | | | | |
| | Intercept | Linear | Square | Cubic | Min | Max |
|---|---|---|---|---|---|---|
| C16_0 | 38.23337 | −12.1674 | 1.247312 | −0.04155 | 8.93 | 11.64 |
| C18_0 | 7.227522 | −7.75059 | 2.04375 | ns | 1.19 | |
| C18_1 | −26254.5 | 1348.859 | −23.0961 | 0.131801 | 59.18 | 57.65 |
| C18_2 | 1058.582 | −151.541 | 7.215724 | −0.11427 | 20.11 | 20.11 |
| C18_3 | −936.244 | 319.6501 | −36.3341 | 1.374954 | 9.17 | 8.5 |

All FAMEs showed a third-order or quadratic relation with PM exhaust. The optimal composition in terms of PM concentration in the exhaust is listed in Table 8. The sum of the different optimal concentrations of the specific FAMEs is 98.58%, which is not 100%. There is a difference of less than 1.5%, caused by the deviation on measurement results.

*3.4. Summary of the Results*

Based upon the polynomial models, we tried to find optimal values for FAME content that lead to a minimal concentration of $NO_x$ and PM in the exhaust. The results are shown in Table 9.

**Table 9.** Overview of the optimal fatty acid composition.

| | NO | $NO_2$ | PM |
|---|---|---|---|
| C16:0 | | 7.03 | 8.93 |
| C18:0 | | 1.63 | 1.19 |
| C18:1 | | 60.32 | 59.18 |
| C18:2 | | 21.78 | 20.11 |
| C18:3 | | 9.24 | 9.17 |
| total | *** | 100 | 98.58 |

*** For NO, no results of interest were found due to the fact that the regression gave only a maximal value for the concentration of NO in the exhaust gases.

## 4. Discussion

This study has explored modelling for the relation between the different FAMEs in biodiesel and the concentrations of NO, $NO_2$ and PM, and searched for an optimal FAME composition that maximally reduces these concentrations in the exhaust gases. In what follows, we will discuss the extent to which the outcome of this model contributes to the overall aim, i.e., to reduce engine emissions, as well as whether the outcome of this model corresponds to literature findings, which may further validate the methodology.

The results of the modelling for NO in the exhaust gases versus fatty acid composition did not permit calculation of an optimal composition in terms of minimal exhaust. However, it has been possible to establish an optimal fatty acid composition for the reduction of $NO_2$ and PM in the exhaust gases using the regression results. Thus, an optimal fatty acid composition for PM as well as for $NO_2$ was found, although no optimum could be calculated to minimize the formation of NO (Table 10).

**Table 10.** Overview of the optimal fatty acid composition in mass percentage.

| RME (m%) | C16:0 | C18:0 | C18:1 | C18:2 | C18:3 |
|----------|-------|-------|-------|-------|-------|
| Mean | 10.70 | 1.775 | 57.91 | 21.06 | 8.60 |
| St. Dev. | 2.36 | 0.21 | 1.22 | 0.95 | 0.51 |
| **RESULTS** | | | | | |
| NO$_2$ | 7.03 | 1.63 | 60.32 | 21.78 | 9.24 |
| PM | 8.93 | 1.19 | 59.18 | 20.11 | 9.17 |

### 4.1. NO$_x$

The influence of biodiesel on the formation of nitric oxides has been widely studied with some remarkable results. Some studies find an increase in NO$_x$ and some find no significant differences, while others even find a slight decrease [1]. In this study, the same type of biodiesel was used for all the measurements, and only specific fatty acid methyl ester concentrations were changed to compare the emissions in order to optimize the fatty acid composition. When changing the composition of the FAME, some other characteristics were changed also, due to the molecular structure, such as, for example, bulk modulus, boiling point, viscosity and cetane number. In general, one can say that cetane number, melting point, combustion heat and viscosity will increase with chain length and saturation [9].

The results of this study show that no optimal values for the fatty acid methyl ester composition was found to have a minimal concentration of NO in the exhaust gases. However, for NO$_2$, an optimal FAME composition was found. A biodiesel should be made with the following FAME composition: 7.03% methyl palmitate, 1.63% methyl stearate, 60.32% methyl oleate, 21.78% methyl linoleate and 9.24% methyl linolenate. It can be seen that, compared to the mean values of the FAME composition, methyl palmitate and methyl stearate (saturated components) show a slightly lower concentration, but the calculated value of methyl stearate stays within its uncertainty range. Methyl oleate and methyl linolenate (unsaturated components) show slightly higher concentrations. Methyl linoleate shows also a slightly higher value, but within the uncertainty range of the mean value.

When considering NO$_x$ in this research, the emphasis lies on NO and NO$_2$. The NO$_x$ in diesel exhaust gases is for the larger part NO and in smaller concentration NO$_2$ [10]. The formation of NO$_x$ depends on three different processes, namely:

(a) Thermal NO$_x$: Above 1500 °C, the reaction of N$_2$ and O$_2$ takes place in the combustion chamber in three steps, known as the Zeldovich mechanism. The reaction kinetics is of the same timescale as the in-cylinder combustion. This is the predominant contribution of NO$_x$ in the exhaust gases.
(b) Prompt NO$_x$: This type is formed under fuel-rich conditions. In [11], an inverse relation between soot and prompt NO$_x$ has been shown.
(c) Fuel NO$_x$: Due to the presence of nitrogen in the fuel, fuel NO$_x$ can be formed. Usually, the concentrations of this type of NO$_x$ can be neglected.

Chang and Van Gerpen [12] reported a decrease in NO$_x$ when more saturated esters are found in the biodiesel. The cause would be an increase in cetane number due to the increase in saturation. This would later be confirmed by a study on cetane number enhancement done by the US EPA [13]. In addition, Li and Gülder found that there was a decrease in nitric acids due to a higher cetane number [14]. Another important feature on the fatty acids in biodiesel is that, according to Rahman et al., fatty acids with shorter carbon chain length also showed less NO$_x$ emission [15], which is consistent with these results. Clearly, the molecular structure of the fatty acids has an influence on the combustion, and thus, the formation of PM and NO$_x$.

Some research proposes that a major reason of the increase of NO$_x$ formation is the change in injection behavior. The needle opens sooner when using biodiesel than when

using mineral diesel due to compressibility and viscosity, which are also functions of the FAME composition [16,17].

However, other hypotheses on the increase of $NO_x$ have been proposed. One reason that has attracted attention is the higher flame temperature and the higher availability of oxygen in the combustion chamber. Ban Weiss et al. [18] showed that unsaturated molecules have a higher flame temperature than saturated molecules. Other arguments on the influence of nitric oxides formation are the fuel spray, again influenced by the viscosity, but also by surface tension and evaporation. Soot formation and the prompt mechanism are also considered as possible reasons of change in $NO_x$ formation. A reduction in soot concentration leads to higher temperature in the combustion chamber, and thus to an increase in thermal $NO_x$ concentration [10].

Zengh et al. [19] mentioned that biodiesel with a comparable cetane number produced even higher concentrations of $NO_x$. Again, the cetane number depends on the molecular composition of the biodiesel, thus on the fatty acid composition of the biodiesel.

### 4.2. Particulate Matter or Soot

In this study, the optimal FAME composition to produce a minimal concentration of PM in the exhaust gases is 8.93% methyl palmitate, 1.19% methyl stearate, 59.18% methyl oleate, 20.11% methyl linoleate and 9.17% methyl linolenate. Again, compared to the mean values of the FAME composition, methyl palmitate and methyl stearate (saturated components) show a slightly lower concentration, but methyl palmitate concentration stays within its uncertainty range. Methyl oleate and methyl linolenate (unsaturated components) show slightly higher concentrations. The concentration of methyl linoleate shows a lesser or no change. The mean value lies lower in value but within the uncertainty range of the measurements. Moreover, it has to be emphasized that the interpretation of the calculated values should be considered carefully, because the total sum of the calculated values, which has to be 100%, shows a deviation of 1.42%.

The use of biodiesel leads in general to a decrease in PM emissions [1]. Even in diesel biodiesel blends, enhancing the biodiesel concentration decreases the particulate matter concentration in the exhaust gases [20]. Various explanations can be given to clarify this decrease. As in the case of $NO_x$ as well as for PM, the molecular structure has an influence on its formation. If carbon chain length increases and unsaturation decreases for the same carbon chain length, there will be an increase in PM emission [15,21]. The results in this study show the same trend, that the unsaturated FAMEs need to be increased to lower the PM concentration in the exhaust gases.

The combustion advance depends on the cetane number of the biodiesel. Where regular diesel has a cetane number of around 48, biodiesel shows a cetane number of around 55 [22]. The higher oxygen content of the biodiesel molecules enables a better combustion, decreasing the need for oxygen in the air–fuel mixture [23].

The content of aromatic molecules is very low or even nonexistent. Aromatic molecules are considered to be soot precursors [12,23]. Moreover, there is no sulphur present in biodiesel, and sulphur is also considered a possible cause of soot formation [24].

Graboski et al. found that when the biodiesel density was lower than 895 kg/m$^3$ and the cetane number lower than 45, the PM concentrations were higher [25]. This means that certain parameters influence each other on the formation of PM, making exact conclusions more complicated.

### 4.3. Chain Length and Saturation

The results in Table 7 show the relative importance of chain length and saturation and we can make conclusions, although with the necessary caution, because chain length changes between C_16 and C_18, so the change in length is limited, because there are only two molecules with different chain length that can be compared.

We have a mean FAME composition which we compare to the optimal composition and then we see that no clear conclusion can be made in function of chain length. In function

of saturation, we can compare four different molecules, C18_0, C18_1, C18_2 and C18_3. According to our results, in order to decrease $NO_x$, we should increase the saturation, and in order to decrease PM, no clear conclusion can be made. Our results in Tables 9 and 10 corroborate literature findings that an increase in chain length correlates with an increase in $NO_x$ and PM according to Pinzi et al. [21] and Rahman et al. [15], whereas Lapuerta et al. [1] showed no clear correlation. Saturation increase leads to $NO_x$ decreases according to all three as well as our data, while for PM no clear conclusion can be drawn.

### 4.4. Further Research Questions

Finally, it should be mentioned that there are a number of mutually non-exclusive methods to reduce engine emissions, apart from the fuel composition. Academia and industry are working on developing innovative diesel combustion systems, able to improve the $CO_2$ emissions, as well as the $NO_x$-Soot trade-offs such as specific bowl design, innovative fuel injection systems, and injection strategy. These technologies could be used also to improve efficiency and performance combined with the alternative fuels in advanced combustion modes such as partially premixed combustion (PPC) and Dual Fuel systems [26,27]. The methods presented here will be useful even in combination with this type of research, as in each instance, optimal fuel compositions will have to be calculated. It may even turn out to be necessary to extend the methodology to include motor management parameters.

### 5. Conclusions

In this study an arbitrary biodiesel was optimized in order to obtain the lowest possible $NO_x$ and PM emission:

- It can be concluded that there exists an optimal fatty acid composition of a biodiesel in order to have the lowest possible $NO_x$ and PM concentrations in the emission.
- There is, as such, a method to calculate the exact composition of the five most common components of the biodiesel, although the immediate results are only applicable on this type of engine.
- A number of factors which influence the formation of $NO_x$ and PM can be contributed to the FAME composition. Among those factors are chain length and saturation.

The results of this study confirm from literature the correlations between both features and formation of PM and $NO_x$.

**Author Contributions:** Conceptualization, R.R.M. and G.P.; methodology, R.R.M.; validation, E.F., R.R.M.; formal analysis, E.F., G.P., R.R.M.; writing—original draft preparation, R.R.M., G.P.; writing—review and editing, F.C.C., S.L., R.V.S.; visualization, E.F.; supervision, S.L., R.V.S.; funding acquisition, R.V.S. All authors have read and agreed to the published version of the manuscript.

**Funding:** This research received no external funding.

**Institutional Review Board Statement:** This research did not include any experimentation on human beings, animals or plant life.

**Informed Consent Statement:** This research did not include any experimentation on human beings, animals or plant life.

**Data Availability Statement:** Data will be kept under embargo for one more year due to possible industrial application.

**Acknowledgments:** The authors wish to thank Hogere Zeevaartschool for providing internal funding and Bioro Ghent for providing biodiesel.

**Conflicts of Interest:** The authors declare no conflict of interest.

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
