# Peer review of "Finding the Optimal Fatty Acid Composition for Biodiesel Improving the Emissions of a One-Cylinder Diesel Generator"

_sustainability, doi:10.3390/su132112089_

Round 1

Reviewer 1 Report

The manuscript deals with an experimental analysis on FAME as fuels for combustion engines to improve the NOx-Soot trade-off. The topic is well aligned with the scope of this journal and as well as with the fuel design towards sustainable mobility. The authors should improve the readability and scientific soundness, the manuscript cannot be accepted in this present form, please carefully revise it to improve the quality, the reviewer has been highlighted several open points below:

Please summarize the title is too long, a maximum of 10-15 words could be sufficient

The authors could extend the introduction discussion reporting that innovative technologies could give a potential boost to the CI engine fuel economy and engine-out emissions reduction. However, they can highlight that the research and industry are working on conventional diesel combustion developing innovative combustion systems able to improve the CO2, and improving the NOx-Soot trade-offs such as specific bowl design, innovative fuel injection systems, and injection strategy. Below are some literature results that could be useful for your discussion: 10.4271/2017-24-0073, 10.4271/03-12-02-0010. These technologies could be used also to improve efficiency and performance combined with the alternative fuels in advanced combustion modes such as PPC and Dual Fuel.

Introduction, please avoid lump sum references, such as XXXXX [1-5], OR 1, 2, 3, 4, 5; all references should be cited with detailed and specific descriptions, and verify the order of the references linked to the text.

Results and discussion, before starting with the analysis of the result please report in two sentences the main contents of this section.

Instead, to report power in kW, the reviewer's suggestion is to report the BMEP [bar], in this way is easier to compare your results with different engines.

Please check the calculation of NOx levels, seems too low order of 10 ppm, Did the authors run in ultra-low NOx regions? How much is the EGR rate?

The reviewer's suggestion is to report the emissions in g/kWh or g/kg_Fuel both NOx and soot.

In the parameter analysis, the authors have not reported information about the emissions and losses (combustion, exhaust, heat transfer, and pumping), an energy balance should be evaluated to optimize the combustion from emissions and efficiencies points of view.

Overall, the manuscript has been well presented. Please emphasize the conclusion section summarizing the main outcomes as bullet points. I would like to recommend it for publication after a revision, please consider to highlights the novelty of this manuscript.

Author Response

REVIEW 1

Please summarize the title is too long, a maximum of 10-15 words could be sufficient

The title has been altered.

The authors could extend the introduction discussion reporting that innovative technologies could give a potential boost to the CI engine fuel economy and engine-out emissions reduction. However, they can highlight that the research and industry are working on conventional diesel combustion developing innovative combustion systems able to improve the CO2, and improving the NOx-Soot trade-offs such as specific bowl design, innovative fuel injection systems, and injection strategy. Below are some literature results that could be useful for your discussion: 10.4271/2017-24-0073, 10.4271/03-12-02-0010. These technologies could be used also to improve efficiency and performance combined with the alternative fuels in advanced combustion modes such as PPC and Dual Fuel.

The reviewer has given a very good and correct remark. However, in this article we were focusing only on a technique to optimize the fatty acid composition. We have done this on a one-cylinder generator with a basic design. We could make some basic adjustments so to clean the fuel reservoir easily and we had a relatively low power so we could easily handle the consumption of the load. Moreover, was the consumption of biofuel low so the cost was limited in this way. Now we have the technique to optimise we started a follow up research with the goals mentioned by the reviewer. We will look into specific injection timing, injection pressure and cooling of the inlet air, and see what the influence is on the fatty acid composition. We did mention the remarks at the end of the discussion.

Introduction, please avoid lump sum references, such as XXXXX [1-5], OR 1, 2, 3, 4, 5; all references should be cited with detailed and specific descriptions, and verify the order of the references linked to the text.

We have surveyed the document for those lumped references. There are three instances where two references have been added in one pair of brackets, and in all these cases, both references support the entire statement to which they belong. The order / numbering has been checked and adapted.

Results and discussion, before starting with the analysis of the result please report in two sentences the main contents of this section.

An introductory phrase has been added to introduce the results and the discussion.

Instead, to report power in kW, the reviewer's suggestion is to report the BMEP [bar], in this way is easier to compare your results with different engines.

Indeed, the reviewer has a point in mentioning that comparing the results for engines by BMEP is far easier. As said in the answer of remark 2, we first of all focused on finding a method to optimise the fatty acid composition. We were not thinking about studying engines or comparing performance of engines. However, we now started a follow up research and then it is useful to start reporting in BMEP because we are expanding the research to diesel engines for barges and maritime transport were the performance of the different types of engines will be compared and then it is useful and even necessary to report in BMEP.

Please check the calculation of NOx levels, seems too low order of 10 ppm, Did the authors run in ultra-low NOx regions? How much is the EGR rate?

Indeed, we reported in low concentrations of NOx and PM. This was done because we diluted the intake of the exhaust gases by a factor of 6,1 which is mentioned in par. 2.1 and 2.3, before the point of measurement. The sensors are meant to measure at low concentrations. 

There was no EGR. We used a very basic diesel generator to measure on and focused on a numerical method to optimize the fatty acid composition.

The reviewer's suggestion is to report the emissions in g/kWh or g/kg_Fuel both NOx and soot.

This is a good remark but in this research the units were chosen in function of the optimisation method. We did not do research on the motor itself or motor management. However, we will use the units proposed by the reviewer in the follow up research because then it is the goal to study the motor and its motor management. The concentrations of NOx and PM are clearly function of the load, but in the method used the effect of the load is regressed out so only the Fatty Acid composition is the only variable left.

In the parameter analysis, the authors have not reported information about the emissions and losses (combustion, exhaust, heat transfer, and pumping), an energy balance should be evaluated to optimize the combustion from emissions and efficiencies points of view.

Also this proposal is  very good in function of the follow up research for the same reason. In this research we only focused on a numerical method to optimize the fatty acid composition, we did not focus on motor or motor management. The next step is indeed focusing on motor management where the energy balance is clearly of importance.

Overall, the manuscript has been well presented. Please emphasize the conclusion section summarizing the main outcomes as bullet points. I would like to recommend it for publication after a revision, please consider to highlights the novelty of this manuscript.

Has been done.

Reviewer 2 Report

1) You have to add references on the fatty acids (line 63 to 64).

2) In Materials and Methods (line 83) please Insert the figure 1 after the end of a paragraph.

Author Response

REVIEW2

1) You have to add references on the fatty acids (line 63 to 64).

Has been done.

2) In Materials and Methods (line 83) please Insert the figure 1 after the end of a paragraph.

The figure has been moved.

Round 2

Reviewer 1 Report

The manuscript has been revised improving quality and readability as well as the scientific content. I would like to suggest it for publication.

Author Response

We thank the reviewer for his helpful comments.